# Self-organized division of cognitive labor

**Edgar Andrade-Lotero** [1]*, **Robert L. Goldstone**[2]

**1** School of Engineering, Science and Technology, Universidad del Rosario, Bogotá, Colombia, **2** Department of Psychological and Brain Sciences and Program in Cognitive Science, Indiana University, Bloomington, Indiana, United States of America

* edgar.andrade@urosario.edu.co

## Abstract

Often members of a group benefit from dividing the group's task into separate components, where each member specializes their role so as to accomplish only one of the components. While this division of labor phenomenon has been observed with respect to both manual and cognitive labor, there is no clear understanding of the cognitive mechanisms allowing for its emergence, especially when there are multiple divisions possible and communication is limited. Indeed, maximization of expected utility often does not differentiate between alternative ways in which individuals could divide labor. We developed an iterative two-person game in which there are multiple ways of dividing labor, but in which it is not possible to explicitly negotiate a division. We implemented the game both as a human experimental task and as a computational model. Our results show that the majority of human dyads can finish the game with an efficient division of labor. Moreover, we fitted our computational model to the behavioral data, which allowed us to explain how the perceived similarity between a player's actions and the task's focal points guided the players' choices from one round to the other, thus bridging the group dynamics and its underlying cognitive process. Potential applications of this model outside cognitive science include the improvement of cooperation in human groups, multi-agent systems, as well as human-robot collaboration.

## Introduction

An individual can often benefit from participating in a group when they can perform just one component of the group's task while other individuals take care of other parts. When this arrangement occurs, we speak of an efficient division of labor. For instance, one study showed that the puzzle of assigning categories to the nodes of a network such that no adjacent nodes have the same category, could be efficiently solved as a collective task if each individual is assigned to a single node and is only concerned with the acceptability of their local sub-network, compared to a situation in which each individual has access to the entire network's configuration [1]; see also [2, 3].

In some collective groups, such as ant colonies or beehives, the division of labor occurs as a genetically designed organization [4–6]. In human groups, on the other hand, the division can

**Data Availability Statement:** The data underlying the results presented in the study are available from (https://github.com/EAndrade-Lotero/SODCL).

**Funding:** EAL was partially supported by the following research awards from Universidad del Rosario: "Becas para Estancias de Docencia e Investigación, 2018" and "Fondo de Capital Semilla, 2018". The funders had no role in study design, data collection and analysis, decision to publish, or preparation of the manuscript.

**Competing interests:** The authors have declared that no competing interests exist.

emerge without leaders or explicit negotiations [7–9]. For example, when a group of individuals has to collectively guess a target number, where the collective guess is the sum of their individual guesses, and the only feedback they receive is for how much their collective guess is greater or lesser than the target, individuals spontaneously differentiate their behaviors to either react or not react to the feedback, and the extent to which role differentiation occurs is predictive of group performance [10].

The division of labor, being a social action, can be studied in one of several ways. A distinctive line of research has focused on viewing social action as arising from the principles of maximization of expected utility [11], which have been formally put together in the game-theoretic approach to social action [12, 13]. When it comes to the division of labor, a game-theoretic approach might represent the situation in the following way. For simplicity's sake, suppose a two-player game where each players' possible strategies consist of sets of task demands and assume only three, say task demands $x$, $y$, $z$. Each particular allocation of task demands to players determines a particular payoff for each player. Say Player *A* performing {$x$, $y$} and Player *B* performing {$z$} affords both players 1 point; but if Player *A* only performs {$x$} and *B* only {$z$} both of them obtain -1 point. By using this approach, a division of labor arises because it constitutes a Nash equilibrium, that is, a division of task demands in which no player can obtain a higher payoff by changing only their allocated task demands—fixing the other players' strategies. [The pair ({$x$, $y$}, {$z$}) is a Nash equilibrium in our example.] However, task demands might be allocated in different ways that are irrelevant to the individuals' payoffs. To come back to our example, it might be irrelevant for *A* to perform either {$x$}, {$x$, $y$} or {$y$, $z$} and it might be irrelevant for *B* to perform either {$y$, $z$}, {$z$} or {$x$}, so that the three pairs ({$x$}, {$y$, $z$}), ({$x$, $y$}, {$z$}) and ({$y$, $z$}, {$x$}) are payoff equivalent. This entails that maximization of expected utility often is not sufficient to explain why individuals act in accord with one particular Nash equilibrium instead of another [14, 15]. Some scholars have suggested that games with multiple non-dominant Nash equilibria are solved on the basis of rough-and-ready rules of thumb, which require only limited knowledge and time. This approach is known as 'bounded rationality' to emphasize that people frequently have memory, attention, and calculation limitations that prevent them from employing perfectly rational strategies [16, 17].

Probably a combination of both approaches—maximization of expected utility and cognitive limitations—is what hits the mark closer, as suggested by Lieder and Griffiths [18]. To bring the point home, consider the following two ways to conceive of the role of focal points in reaching an equilibrium. According to Schelling [19], strategies might be ordered—albeit in a payoff-irrelevant way—by the labels with which they are represented by the subject. Such label ordering induces an ordering in the set of all Nash equilibria, which thereby might be reduced to the highest ranking point in that order. A focal point can be said to guide an individual's choice in different ways. On the one hand, a focal point can provide reasons why it would be rational to choose a focal point: because the individual thinks that the other individual would think that the other individual would think that . . .that the other individual would be drawn towards that particular choice [20–22]. On the other hand, a focal point might simply tend to draw an individual's attention to go for a particular choice instead of another. The latter approach suggests a cognitive bias which often helps the individual maximize their expected utility without performing involved calculations. We believe that this way of conceiving of focal points—heuristics for optimization with limited cognitive resources—is more appropriate and that there must be fast-and-frugal heuristics that individuals can use to tacitly negotiate a division of labor through an iterative process [23].

Besides Schelling's points, another antecedent for a coordinating heuristics in an iterative scenario is that one individual stays put—repeating their choice every round—while the other

adapts their action to match that of the former (it has been formally proven that this is the most efficient way to coordinate actions in an iterative process [24]). However, we still require an account as to how players tacitly negotiate who is to stay put and who is to adapt their action to achieve coordination.

We have developed an iterative two-person game in which there are multiple ways of dividing labor, but in which it is not possible to explicitly negotiate a division. Our methodology consisted in the implementation of this game both as a human experimental task and as a computational cognitive model. Our results were in accordance with the findings mentioned earlier, but we were able to go one step further. We found evidence that the perceived *similarity* between a player's actions and the task's focal points guided the players' choices from one round to the other. By studying a formal model of this heuristics and fitting it to our behavioral data, we were able to propose a bridge between the group dynamics and its underlying cognitive process.

Some potential applications of our model outside cognitive science include the improvement of cooperation in human groups [25] and multi-agent systems [26], as well as human-robot collaboration [27]. Possible suggestions are presented in the Discussion section.

## Materials and methods

### Participants and procedure

Participants were 90 undergraduate students from the standing human subject pool at Indiana University in Bloomington. This pool, which does not contain legal minor participants, consists of students from introductory psychology courses (P101 and P102) who could participate in four experiments in a semester to satisfy their course requirement to engage in hands-on experiences with contemporary human behavioral experimentation. Participation in our study satisfied one of these four experiments. All participants in our study had to read an informed consent form and give their agreement before they could participate in the study. Participants were run in 10 experimental sessions, each one requiring an even number of participants to be grouped into dyads. If an odd number of participants turned up to the session, one of them was randomly chosen and sent home. The numbers of dyads in each session were as follows: 4, 5, 3, 6, 4, 2, 6, 3, 8, and 4. Participants sat in a university computer lab, each at a sound- and sight-isolated personal computer running a version of the game implemented in the node-Game platform [28]. The computer randomly paired participants into dyads and each dyad participated in 60 rounds of the game. Participants were instructed not to talk to each other and were not informed about who was paired with whom.

### The task

The task is a two-player game in which players interact with 64 tiles arranged in an $8 \times 8$ grid (see the left panel in Fig 1). The grid can either hide a unicorn beneath one of the tiles or else it can be absent from the grid. Either event can occur with equal probability. At the beginning of each round, the computer chooses whether or not there is a unicorn, and if there is one, it randomly chooses a tile—each one having an equal probability of being chosen—and places the unicorn beneath it. Then, players have to guess whether there is a unicorn or if it is absent from the grid by uncovering tiles one at a time, with both players uncovering tiles simultaneously, in order to see what lies beneath them. What tiles have been uncovered and whether there is or not a unicorn is only known to the player that uncovers these tiles. However, in the event that both players uncover the same tile, it changes its color and both players can immediately see this. At any time during the round, each player can make a guess as to whether the

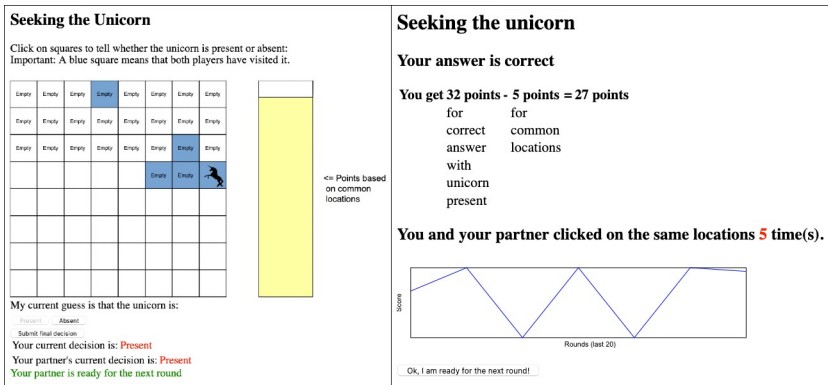

**Fig 1. The experimental task.** The left panel shows the grid as displayed to each player. By uncovering a tile, the player knows whether it is empty or contains the unicorn. Such information is private for the player. Tiles uncovered by both players have a blue background and both can immediately see this coloration. They also have access to each other's guesses about whether the unicorn is present or absent. The yellow column on the right decreases as the number of overlapping tiles increases. The round ends when both players submit their final decision. In the right panel we show the screen displaying the score and the score history over the last 20 rounds.

unicorn is present or absent from the grid. The other player will know this information and they can use it to inform their own guess. The round ends when both players announce that their guess is a final decision, and then they are shown their individual scores from the round (see the right panel in Fig 1). The score depends on whether the player's guess is correct (32 points) or incorrect (-64 points), subtracting the number of tiles that were uncovered by both players. Note that if a player guesses "Absent" while the other guesses "Present", they will receive different scores.

There are a number of characteristics of the task that are worth noting. Conveniently for our purposes, the division of labor is represented as a split of the grid, since each tile represents a task demand. Needless to say, there are a large number of these and the reason for this complexity is twofold. Earlier pilot runs with a $4 \times 4$ grid showed that players were not keen to split it and that they were more than happy to uncover all tiles regardless of the negative payoffs. We also want to include an incremental way of dividing labor because this is what allows us to tell whether one player is adapting to the apparent region being selected by the other player or not. Observe that each player can monitor the other player's actions, but only to the extent that they overlap with each other. Therefore, players still have to try and figure out the other player's strategy, and these strategies evolve over rounds of interaction. Finally, although players cannot explicitly communicate with each other, their guesses are public knowledge, which is important for facilitating coordinated action. When a player constrains their search to a particular area, they still have to rely on the other player's report on whether there is or not a unicorn in the complementary area.

To bring the presentation of our task to a close, we would like to mention that it can easily model collaborative search tasks (cf. [29]) where participants must efficiently divide the visual field, such as two bodyguards splitting the range of vision to scan for potential threats; or two kids looking at a picture book (e.g., *Where's Wally?*); but it can also accommodate more abstract division of labor tasks by considering that each tile represents a task demand and each region in the grid a set of task demands (so called "wicked problems" are obviously outside the scope of this model since they do not afford a division into separate task demands because of their complex and interconnected nature).

## Measures

The following measure, which we call the Division of Labor Index (DLINDEX), determines the extent to which players split the grid into complementary regions:

$$\text{DLINDEX} = \frac{\text{Tiles uncovered by one or both of the players} - \text{Overlapping tiles}}{\text{Tiles in the grid}}$$

This measure instantiates the intuition that it is beneficial if a dyad collectively uncovers all of the tiles (first term) and does not overlap on any tiles uncovered (second term). Observe that it ranges from 0 to 1 with 1 being ideal division of labor and 0 being least efficient. Ideal performance is achieved when both players uncover the entire grid and do not overlap at all.

We also define the similarity between two regions $a$ and $b$ in the grid in the following way:

$$\text{sim}(a, b) = \frac{\text{Number of tiles in both } a \text{ and } b}{\text{Number of tiles in } a \text{ or } b} \tag{1}$$

In case both $a$ and $b$ are empty, $\text{sim}(a, b) = 1$. Additionally, we measure how consistently a player uncovers tiles from one round to the next:

$$\text{Consistency}_n = \frac{\text{Overlapping uncovered tiles from Round } n-1 \text{ to Round } n}{\text{Tiles uncovered in either of the two rounds}}$$

In case a player uncovers no tiles on the two rounds, $\text{Consistency}_n = 1$. Observe that this measure ranges from 0 to 1 with 1 meaning that the player uncovers the same tiles on both rounds, and 0 meaning that the player uncovers a completely different set of tiles from one round to the next.

## Results

To begin with, observe that rounds on which the unicorn was *present* provide us only with partial information as to how players split the grid. On these rounds, players did not have to uncover every tile because when they found the unicorn, they responded that the unicorn was present and finished the round. For most of our analysis this represents a problem, since we are precisely looking at how players split the grid. But one thing that is interesting to look at with respect to rounds with unicorn present is how they influenced players' behavior on the subsequent round (with unicorn absent). We observed that such rounds made players more active in that they uncovered more tiles on the next round. The average number of uncovered tiles in a round with unicorn absent is around 34 (SD = 19.3) when the unicorn was present in the previous round, and only 31 (SD = 18.8) when the unicorn was absent ($t(1177) = 3.417$; $p < 0.001$). For the ensuing analyses, unless explicitly stated otherwise, we will only consider data obtained during rounds on which the unicorn was *absent*.

In Fig 2 we show the results of four representative dyads. We measured the dynamics of dyads in terms of each player's consistency and score, and a grid with magnitude-coded tiles (the darker the tile, the more times it was selected by the player throughout the experiment). We can also observe the dyad's DLINDEX through the rounds. Three of these dyads (top left, top right and bottom left quadrants) converged on a split of the grid, which can easily be seen in the magnitude-coded tiles, as well as in the high values of all measures. The remaining dyad (bottom right quadrant) exemplifies an unsuccessful dynamics.

As for the overall performance of all dyads, we classified them into eight cluster regions, which are shown in Fig 3. This classification was obtained on the basis of a visual inspection of

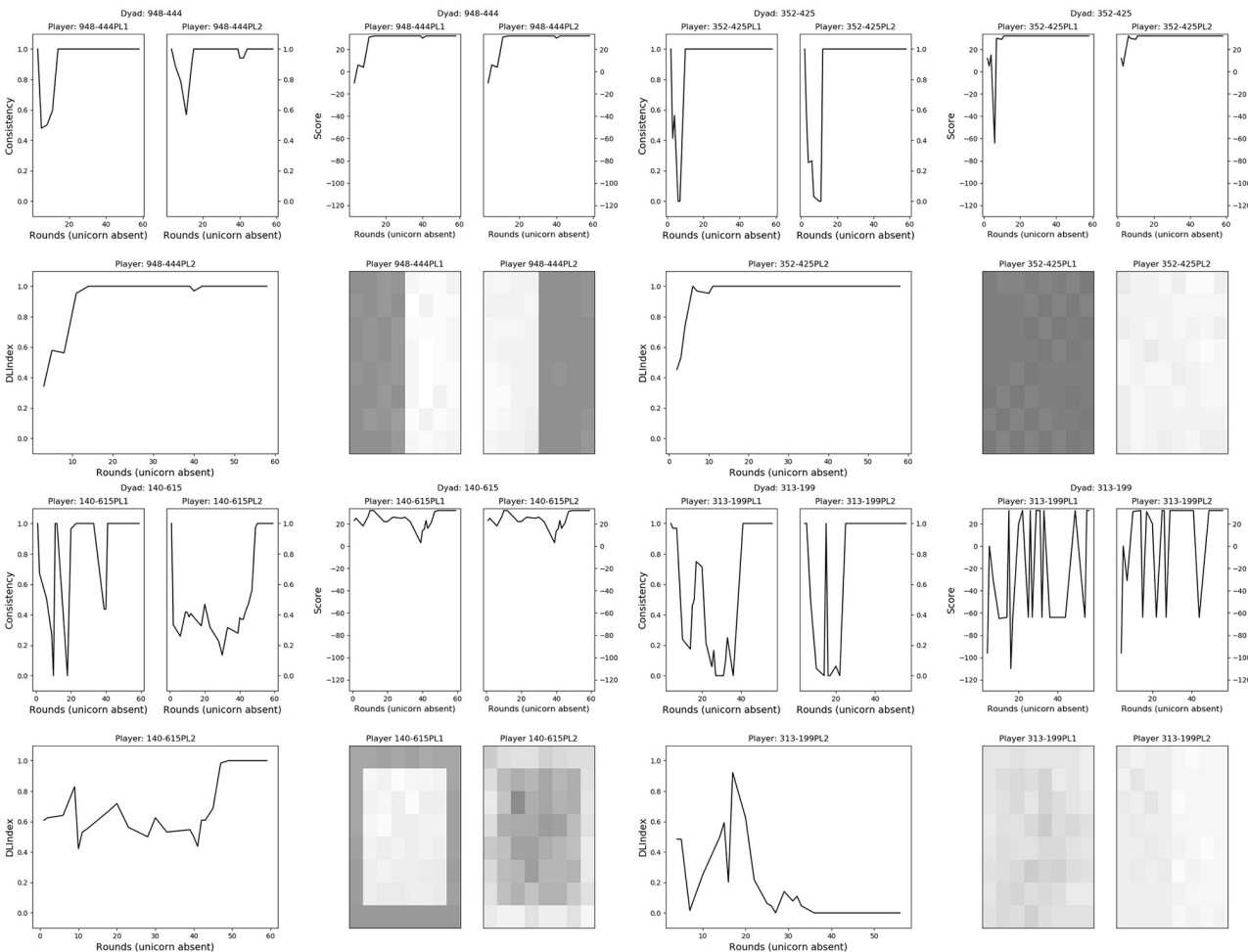

**Fig 2. Examples of observed behaviors.** Each quadrant contains four (sets of) panels displaying the dynamics through 60 rounds of gameplay for a dyad. The dynamics is measured by means of consistency (top left, within each quadrant), score (top right), and magnitude-coded tiles—i.e., how often a particular tile was selected—from each player (bottom right), as well as the dyad's DLINDEX (bottom left). Only rounds on which the unicorn was absent were considered. The top-left quadrant presents a dyad that relatively quickly converged on a LEFT-RIGHT split. The top-right quadrant presents another quickly convergent dyad, but this time on an ALL-NOTHING split. The bottom-left quadrant shows a slowly convergent dyad that negotiated an INSIDE-OUTSIDE split, and the bottom-right quadrant shows a dyad with unsuccessful division of labor.

the magnitude-coded tiles and a semi-automatic classification of regions. Our results show that there were only four stable, successful pairs of complementary regions in the grid: the LEFT-RIGHT, TOP-BOTTOM, ALL-NOTHING, and INSIDE-OUTSIDE splits. We call them the *focal splits*. Only dyads creating focal splits obtained an above-average DLINDEX, except for one dyad with no discernible stable region that nevertheless has an average DLINDEX of 0.82 (this dyad determined the Mix type of split in Fig 3). We conclude that 26 out of 45 dyads successfully split the grid. This represents over 57% success in self-organizing division of labor.

If our paradigm were a task in which players had to converge on a split of the grid on a single round, our data show that the average DLINDEX would be close to 0.36 (SD = 0.32). By comparison, in our iterated task, the average DLINDEX rose to 0.738 on Round 60 (SD = 0.34). The difference between these averages is statistically significant (t(46) = 4.09, p < 0.001; see also top right panel in Fig 4). Perhaps not surprisingly, since there are many focal splits on which players could converge, this shows that an efficient division of labor does not occur on the first round, and that the iterated nature of our task facilitates its emergence. On the first round of

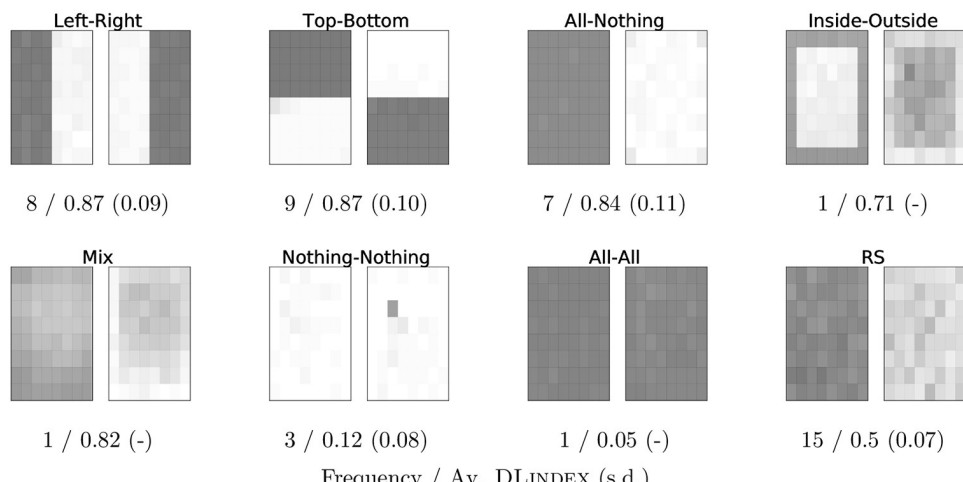

Frequency / Av. DLINDEX (s.d.)

**Fig 3. The eight types of splits of the grid that could be observed from our data.** Each panel shows two grids, one for each player, with the regions uncovered through 60 rounds (on unicorn absent trials). The darker the tile, the more times it was uncovered by the player across all rounds. For each type we also show the observed frequency and the average DLINDEX with its standard deviation.

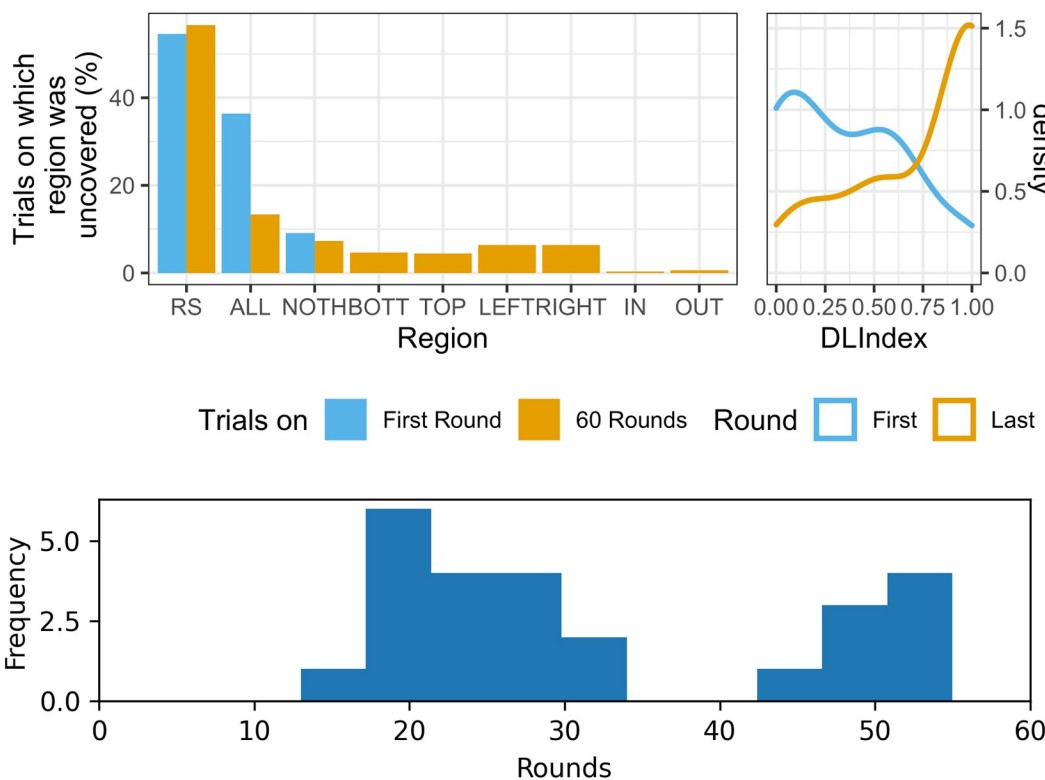

**Fig 4. Convergence on a successful division of labor.** Top left panel: percentage of trials on which each type of region was uncovered (first round $N = 22$; all rounds $N = 1244$). Top right panel: kernel density estimate of DLINDEX (first round $N = 22$; last round $N = 30$). Bottom panel: histogram of first round on which DLINDEX reached a stable high value (by considering the first round on which the rolling mean (window = 2) of the DLINDEX was above 0.995). Only successful dyads were considered ($N = 26$).

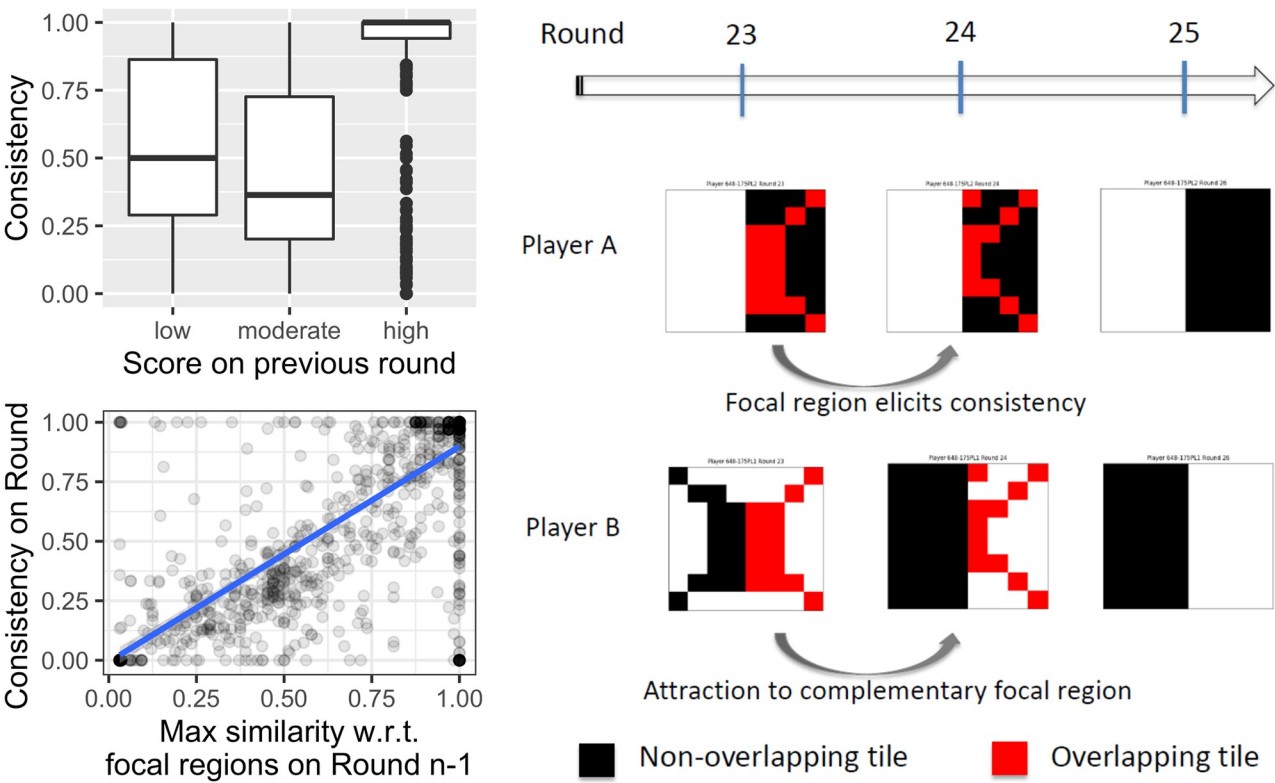

**Fig 5. Evidence of bounded-rationality heuristics.** Top left panel: Boxplot of consistency for three different levels of score on the previous round. Bottom left panel: Regression of consistency with respect to maximum similarity to a focal region on the previous round. In both panels, each data point represents a trial ($N = 1134$). Right panel: Transition from one round to the next, taken from an actual gameplay. In each grid, black tiles represent tiles uncovered by one player and red tiles were uncovered by both players. Player A's transition illustrates 'stubbornness' and Player B's illustrates the attraction exerted by the complement of A's focal region, which is also a focal region. See details in the text.

the game, no player uncovered regions BOTTOM, TOP, LEFT, RIGHT, IN, or OUT (see top left panel in Fig 4) and it is not until Round 4 that one of these regions can be observed. Moreover, it took at least 11 rounds for dyads to converge on a stable division of labor and the majority of successful dyads converged around the 20th round, as shown in the histogram in the bottom panel of Fig 4. To obtain this histogram, we inspected the successful dyads and determined the first round on which the rolling mean (window = 2) of the dyad's DLINDEX reached a value above 0.995, which can be interpreted as the moment the dyad converged on a stable division of labor.

But how did these divisions of labor emerge? We observed that, in general, dyads moved from lower to higher levels of DLINDEX, and players in a poorly performing dyad tended to more frequently change their tile selection strategy from one round to the next compared to players in a well performing dyad. Moreover, we found a positive correlation between a player's consistency on Round $n$ and their score on Round $n − 1$ ($r(1132) = 0.23$, $p < 0.001$). In order to take a closer look at this correlation, we cut the score on Round $n − 1$ into three intervals: low (from -128 to 15), moderate (from 16 to 28), and high (from 29 to 32). The boxplots of these intervals with respect to consistency on Round $n$ can be seen in the top left panel in Fig 5. Here we can see that, despite a great deal of outliers, more often than not high scores have a higher degree of consistent behavior on the next round. We also cut consistency on Round $n$ into three intervals: inconsistent (from 0 to 0.31), moderately consistent (from 0.31 to 0.86), and consistent (from 0.86 to 1). The cross-frequencies can be found in contingency

**Table 1. Cross-frequencies of score on round $n - 1$ and consistency on round $n$.**

| Score on Round $n - 1$ | Consistency on Round $n$ | | |
|---|---|---|---|
| | Inconsistent | Moderately consistent | Consistent |
| Low | 81 | 144 | 75 |
| Moderate | 117 | 109 | 44 |
| High | 84 | 32 | 448 |

Table 1, from which we can confirm via a chi-square test that these variables are positively related ($\chi^2(4, N = 1134) = 429$, $p < 0.001$). We conducted an ordinal logistic regression analysis to investigate this relation. Low values of the predictor variable, score on Round $n - 1$, were found to favor an inverse relationship to consistency on Round $n$ as compared to moderate scores (orderedlog $-$ odds(Estimate) $= -0.54$, SE $= 0.15$, $p < 0.001$), but they favor a positive relationship as compared to high scores (orderedlog $-$ odds(Estimate) $= 1.98$, SE $= 0.15$, $p < 0.001$). The odds increase in consistency from moderate to high scores is about 7 times larger than the odds increase in consistency from moderate to low scores. This supports the hypothesis that players used, at least to some extent, a "Win Stay, Lose Shift" heuristic.

However, this does not account for all the characteristics of the dyadic interaction. When we predict DLINDEX as a function of consistency, we see that, perhaps not surprisingly, dyads consisting of individuals who are relatively consistent in their tile selection strategies tend to divide labor better ($\beta \approx 0.36$; $p < 0.001$). Moreover, we also observed an interaction such that dyads with players that differ in their consistencies tend to divide labor better than predicted when players have a large amount of overlap in their selected tiles. That is, if both players overlap considerably in their selected tiles, the best division of labor is obtained when one player is consistent and the other player is not. The evidence for this claim comes from comparing the linear regression model above with a model that includes the interaction between, on the one hand, the absolute difference in consistency between players on a given round and, on the other hand, the number of overlapping tiles on the previous round:

$$\text{DLINDEX}(n) \sim \alpha + \beta_1 * \text{Consistency}(n) + \beta_2 * \text{difConsist}(n)$$
$$+ \beta_3 * \text{Overlap}(n - 1) + \beta_4 * \text{difConsist}(n) * \text{Overlap}(n - 1)$$

(2)

Our data show that this interaction is positive ($\beta_4 \approx 0.01$; $p < 0.001$; the interaction is also positive and significant when the contribution of Consistency($n$) is not included). Moreover, an analysis of variance test ($f(2) = 40.6$, $p < 0.001$) confirms that this interaction effect accounts for significantly more variance in performance compared to the main effects. These results indicate that dyads eventually tend to most effectively divide labor despite initially overlapping in their tiles when one player is consistent/stubborn and the other player is inconsistent/flexible, giving rise to complementary degrees of reactivity to overlap [10].

The relative success of dyads that have complementary levels of stubbornness raises the question: what predicts whether a player will become stubborn? We found that if a player tends to select tiles consistent with a focal region (that is, one half of a focal split), they tend to be more consistent. In other words, the more similar a player's tile selection strategy is to a focal region, the more stable their selections become, likely because they believe that they are forming one half of a viable division of labor. The regression model of consistency with respect to maximum similarity to a focal region confirms this effect ($\beta = 0.9$; $p < 0.001$; see bottom left panel in Fig 5). How, then, does the other player determine that they have to select tiles in the appropriate complementary region, given that a player only has access to their own uncovered tiles and not the other player's uncovered tiles? Most likely, this is achieved because players

have access to overlapping tiles, from which the other player's selected tiles can be inferred with reasonably high validity.

One mechanism that accounts for some players' shifts in selected tiles is based on a combination between stubbornness and the similarity between a focal region and the overlapping tiles. If one player's overlapping selected tiles are sufficiently close to a focal region, this can be used as a signal for the other player to select the corresponding, complementary focal region. In Fig 5, right panel, we take a close look at an actual gameplay from a dyad in which this mechanism is prominent, as exhibited by Player B's transition. On Round 23 the overlapping tiles are similar to the focal region RIGHT, which induces B to select every single tile in the complementary LEFT region. Observe that B not only re-selected the left region's tiles from the previous round, but uncovered the entire LEFT region. The revealed overlapping tiles are the same for both players, so Player A's attention is also attracted by LEFT. Nevertheless, given that A has uncovered the focal region RIGHT, they tend to become "stubborn" in the sense of resisting substantial change to their uncovered tiles. The combination of attraction towards a focal region by the player close to it and attraction towards its complement by the other player represents a decision process that we call the Focal Regions as Attractors heuristic (FRA). To be sure, the process is often more gradual than shown in Fig 5, and there certainly are other factors at play. However, even the simplest form of this heuristic is evident in our results.

## Computational models

### Three models for seeking the unicorn

We put our previous explanations to the test by devising computational models instantiating our heuristics, and then fitting them to our behavioral data. We consider three models. The first one, MBIASES, instantiates the idea that some regions are more visually salient than others, so that players randomly choose a region on the basis of biases shared by all participants. These biases favor focal regions. The second model, WSLS, is an implementation of "Win Stay, Lose Shift", which builds upon the former model and also implements the condition that a sufficiently good score leads to a re-selection of the previous region (if it was a focal region). The third model, FRA, uses the two previous heuristics plus one we call "Focal Regions as Attractors."

The rationale behind our models is as follows: A model defines the probability that each region $k$ be selected as the one to be explored on the next round. We divide the collection of all regions into nine categories: The eight focal regions plus a type of region, RS (Random Search), which stands for all other regions in the grid. [If a player chooses RS, then the player chooses a random region in the grid in which all tiles have equal probability of being chosen.] Furthermore, to determine the probability of $k$, a model uses an attract($k$) function, which represents the extent to which a player is inclined to choose region $k$, given the current state of the game (what this state amounts to and how the attract($k$) function is defined depends on the model at hand). The probability is then determined by the following formula:

$$P(k) = \frac{\text{attract}(k)}{\sum_{r \in \mathcal{K}} \text{attract}(r)} \tag{3}$$

Starting with MBIASES, the attract($k$) function does not depend on the current state of the game, only incorporating the psychological salience of region $k$, bias$_k$. This term represents how inclined the player feels toward $k$, all other things being equal, and is expected to be higher for pre-experimentally salient regions. We assume that the sum of these biases is 1 and, therefore, $P(k) = \text{attract}(k) = \text{bias}_k$.

As for WSLS, suppose that on round $n$ the player uncovered the region BOTTOM. If the score on this round was good enough, WSLS increases the attractiveness of BOTTOM. This effect is achieved by means of a threshold function multiplied by a factor $\alpha$ that increases this region's attractiveness for the player.

In this model, we assume that the current state of the game is represented by the vector ($i$, $s$), where $i$ is the region explored on the previous round and $s$ the obtained score. The attract function is defined in the following way:

$$\text{attract}(k, i, s) = \text{bias}_k + \alpha * \text{thresh}(s, \beta, \gamma) * I(k, i) \tag{4}$$

The second term contains the functions thresh and I, defined as follows:

$$\text{thresh}(s, \beta, \gamma) = \frac{1}{1 + e^{-\beta(s-\gamma)}}, \qquad I(k, i) = \begin{cases} 1, & \text{if } i = k \neq \text{RS} \\ 0, & \text{otherwise} \end{cases}$$

Here, $s$ is the score, which takes a value between -128 and 32. The function $thresh(s, \beta, \gamma)$ has an S shape and takes values in the open interval (0,1). It goes from values near 0 when $s$ is lower than $\gamma$ to values near 1 when $s$ is higher than $\gamma$; the steepness of this transition is determined by $\beta$. The parameter $\alpha$ determines the extent to which the score increases the player's tendency to choose $k$, when the score is higher than $\gamma$. For the sake of simplicity, the parameter $\beta$ is not taken to be a free parameter in Eq 4. It is assumed to have a constant value determining a high steepness of the threshold function (we have set $\beta = 30$ for our simulations and parameter fit). The effect of $I(k, i)$ in Eq 4 is that the only region that has its bias modified is region $i$ (i.e., the region explored on the previous round) and only if this region is a focal region. The value of attract($k$) for the remaining regions is equal to $bias_k$, because $I(k, i) = 0$ when $i \neq k$.

As for FRA, this model extends the previous one by considering a combination of two mechanisms: how focal regions attract a player's attention, and how the overlapping region shifts a player's attention away from it and towards the complement of the focal region that is similar to this overlap. By means of example, suppose that on round $n$ Player 1 uncovered tiles as shown in the top left panel in Fig 6; call this region $i$. The attraction mechanism considers the similarity between $i$ and each focal region, increasing the probability of choosing a focal region on the next round as $i$ increasingly resembles it. In our example, $i$ is most similar to region BOTTOM. Now, the repulsion mechanism considers the overlapping region (red tiles in the figure); call it $j$. This mechanism measures the similarity between $j$ and the complement of each focal region. In other words, the more similar $j$ and $k$ are, the more attractive *the complement* of $k$ becomes. In our example, the overlapping region is most similar to both BOTTOM and LEFT, so the repulsion mechanism shifts attention towards TOP and RIGHT. Finally, the FRASim measure adds up the two previous measures. Observe that, in our example, FRASim with respect to RIGHT is higher than the rest, and if this value overcomes a given threshold, the attractiveness of RIGHT increases substantially.

More formally, we assume that the current state of the game is represented by the vector ($i$, $s$, $j$), where $i$ is the region explored on the previous round, $s$ is the obtained score, and $j$ the region formed by the overlapping tiles. The attractiveness of $k$ is defined in the following way (see Fig 6 box (b) for an illustration of the surfaces determined by this equation):

$$\begin{aligned} \text{attract}(k, i, j, s) = \quad & \text{bias}_k + \alpha * \text{thresh}(s_n, \beta, \gamma) * I(k, i) \\ & + \delta * \text{thresh}(\text{FRAsim}(i, j, k), \epsilon, \zeta) \end{aligned} \tag{5}$$

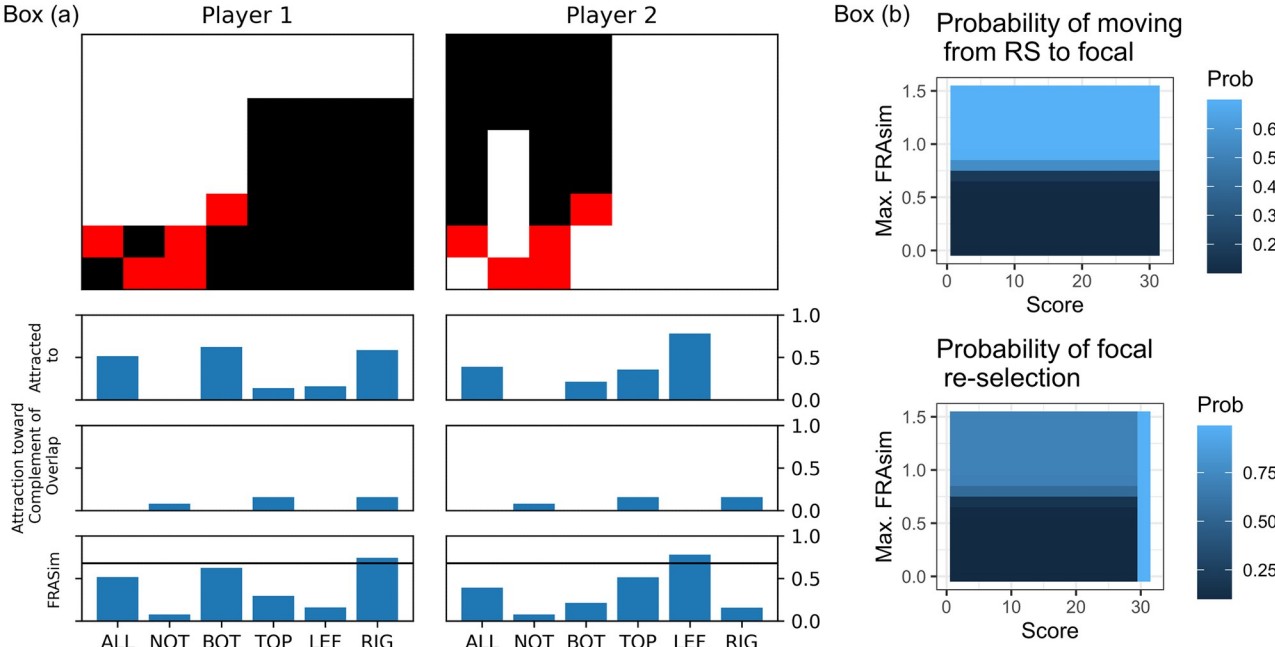

**Fig 6. The mechanisms of FRA in action.** Box (a) top panels: Regions explored by each of the two players with overlapping tiles in red, drawn over an actual gameplay. Box (a) second from top panels: Attraction exerted by focal regions in terms of their similarity to the explored region. Box (a) third from top panels: Attraction towards a focal region in terms of the similarity between the complement of the overlapping region and the focal region. Box (a) bottom panels: FRASim of each focal region, which adds up the two previous similarities. These panels also display a threshold line which, when a region surpasses it, triggers considerable attraction towards this region (focal regions IN and OUT are not shown for ease of presentation). Box (b) top panel: Heatmap of the probability surface representing the probability that a player who has selected a random region on the previous round chooses a focal region, as a function of score and FRASim levels. In this case, FRA predicts that the score level does not influence this movement, which can be reflected by the uniform surface along the score-axis. Box (b) bottom panel: the surface corresponding to the probability of focal re-selection depending on the previous score and the maximum FRASim obtained on the previous round. Observe that, in this case, high score levels predict high probability of re-selection.

Observe that the first two terms in Eq 5 are the same as in Eq 4. The third one is new. Here, the function $FRAsim(i, j, k)$ is defined in the following way:

$$\text{FRAsim}(i, j, k) = \text{sim}(i, k) * \text{Focal}(k) + \text{sim}(j, \bar{k}) * \text{Focal}'(k)$$

The function sim was defined in Eq 1. The FRASim function returns the sum of two terms, the first is the similarity between region $i$ (the region the player just uncovered) and focal region $k$; and the second is the similarity between the overlapping region $j$ and the complement of region $k$ ($\bar{k}$). This only occurs when $k$ is a focal region and is different from ALL. [The requirement that $k$ be different from ALL is due to the following reason. When considering a situation in which there is a high similarity between $j$ and NOTHING, we do not want to increase the attractiveness of ALL, because an empty overlap could mean that players have converged on a split of the grid.] The desired effect is obtained by multiplying by $Focal(k)$ and $Focal'(k)$, which are defined as follows:

$$\text{Focal}(k) = \begin{cases} 1, & \text{if } k \neq \text{RS} \\ 0, & \text{otherwise} \end{cases} \qquad \text{Focal}'(k) = \begin{cases} 1, & \text{if } k \neq \text{RS and } k \neq \text{ALL} \\ 0, & \text{otherwise} \end{cases}$$

The parameter $\delta$ in Eq 5 determines the extent to which $FRASim(i, j, k)$ modifies attract(k). The parameter $\epsilon$ determines the steepness of the thresh function and $\zeta$ determines the threshold. For the sake of simplicity, parameters $\beta$ and $\epsilon$ are not taken to be free parameters in Eq 5.

They are assumed to have a constant value determining a high steepness of both threshold functions (we have set $\beta = \epsilon = 30$ for our simulations and parameter fit). Note that the extra parameters from FRA with respect to WSLS are $\delta, \epsilon, \zeta$. Moreover, Eq 4 can be obtained from Eq 5 when $\delta = \epsilon = \zeta = 0$. That is, WSLS is a nested, restricted model within FRA. Observe also that MBIASES is a nested, restricted model within WSLS and FRA.

## Simulations

We have simulated a dyad's behavior in the following way. At the beginning of the game, each player randomly chooses a region in the grid, in the form of a list of squares. The probability of choosing the initial region is given by the biases determined by the parameters of the model. Then, players simultaneously check the first tile in their list. If either player finds the unicorn, they issue "Present" and declare that their decision is final. The other player has access to this information and also issues "Present" and declares that their decision is final, thus ending the round. If not, both players continue to check the next tile on their list until one of them finds the unicorn or until their respective lists are exhausted. When a player exhausts their list, they issue "Absent" and declare that their decision is final. If both players declare that their decision is final, the round ends. At the end of the round, each score is calculated and players are informed of the overlapping area during the round. With this information, each player chooses which region to visit on the next round by means of the probability function in Eq 3.

Simulations allow us to check that our models display the desired qualitative behavior and, since they are nested—building one mechanism on top of another—we can see how each set of extra parameters modifies the individual-level behavior. Additionally, we can see how modifications at this level affect the behavior at the dyadic level.

For each model, we simulated 150 dyads, each playing 60 rounds of the game. Results are summarized in Fig 7. In the top left panel we can see the frequency of each focal region, as they were explored by a player on a given round—we call this a *trial*. That is, for instance, if we consider trials from MBIASES, we can observe that in 10% of the cases players chose region BOTTOM. This corresponds to the parameter bias$_{\text{BOTTOM}}$, which was set to 0.1 (see Table 2). For the model in question, all frequencies correspond to their respective model parameter. [It is important to note that we reduced the number of parameters by considering that bias$_{\text{LEFT}}$=bias$_{\text{RIGHT}}$=bias$_{\text{BOTTOM}}$=bias$_{\text{TOP}}$, and that bias$_{\text{IN}}$=bias$_{\text{OUT}}$. This is justified not only because we are taking pairs of complementary focal regions, but also because of the observed frequencies in our human dataset (see Fig 3).] This means that the qualitative behavior of these parameters is as expected: they determine, all other things being equal, the probability that a player chooses the respective focal region. Next, consider the contribution of parameters $\alpha, \beta$ and $\gamma$, which determine the "Win Stay, Lose Shift" mechanism. This can be appreciated in the top right panel in Fig 7, in which we present the boxplot of the players' consistency on Round $n$ for three different levels of score obtained on Round $n - 1$. Observe that for WSLS and FRA, high scores have a higher degree of consistent behavior on the next round, which represents the players' tendency to re-select tiles from rounds on which their score was higher. This was not so for MBIASES, since it does not implement the mechanism in question. Furthermore, observe that "Win Stay, Lose Shift" has a clear effect at the dyadic level, since we can see that players converged on a successful split of the grid, as shown in the bottom right panel in Fig 7. This panel shows the average over 150 dyads of the DLINDEX per round. Only WSLS and FRA show a convergence towards high values of DLINDEX. The FRA model's better performance on earlier rounds is explained by the extra mechanism it implements. The mechanism depends on parameters $\delta, \epsilon, \zeta$ which control the influence of

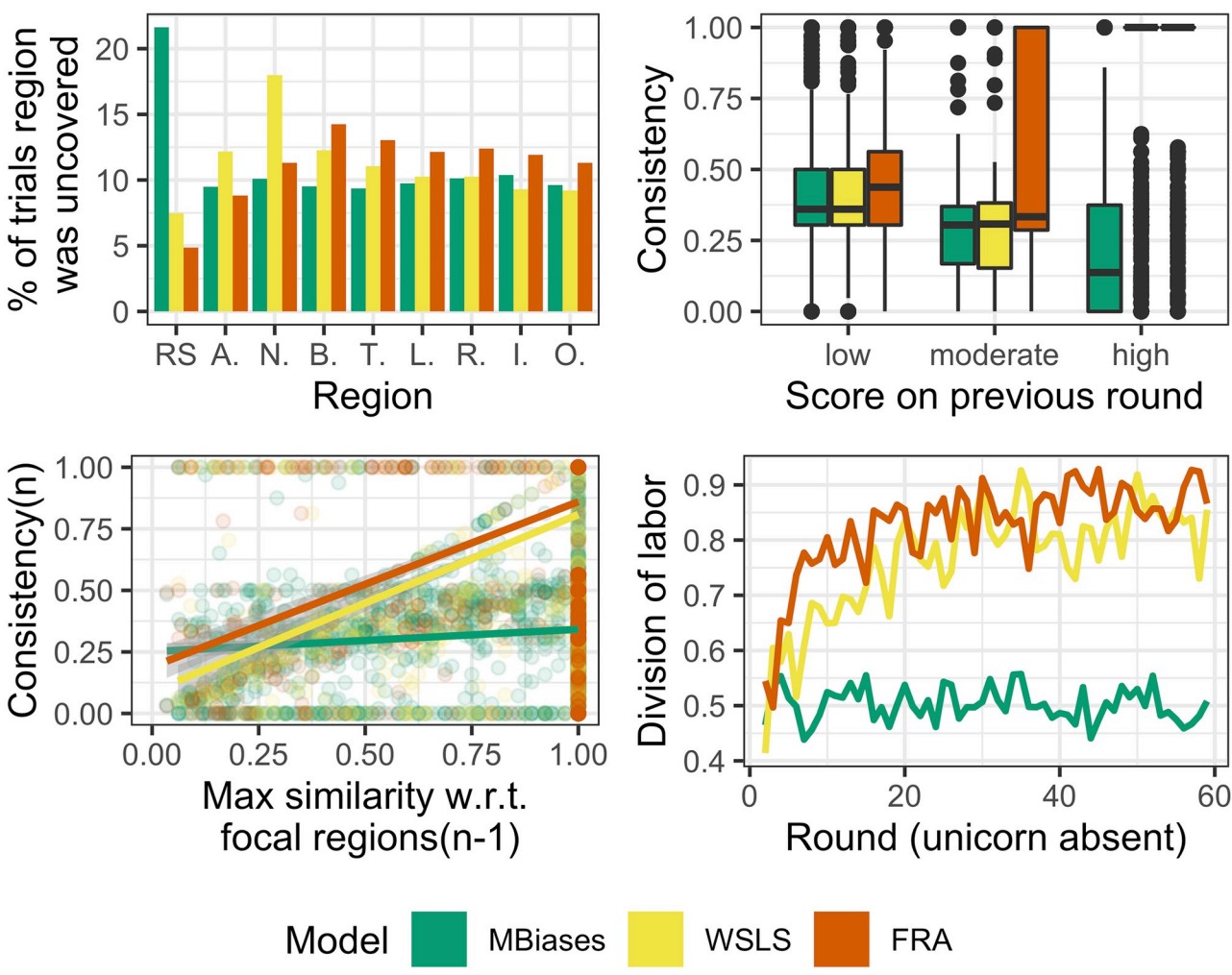

**Fig 7. Simulations results.** Top left: Percentage of trials on which a region was chosen and uncovered. This panel shows that, in MBIASES, the parameter bias$_k$ determines the frequency of trials on which region $k$ is selected. Top right: Evidence of "Win Stay, Lose Shift" heuristic, which is present in both WSLS and FRA but not in MBIASES. The panel shows the boxplot of the players' consistency on Round $n$ for three different levels of score obtained on Round $n - 1$. Bottom left: Evidence of how similarity to focal regions elicits stubbornness. This mechanism is not present in MBIASES, which is reflected by the almost horizontal line in the plot. The effect of this heuristic is quite clear in FRA and WSLS. Bottom right: Qualitative behavior in terms of DLINDEX per round, averaged over 150 dyads. We can see that, while MBIASES doesn't take off, WSLS and FRA can be quite successful in negotiating a split of the grid. The latter is more efficient than the former, due to the extra heuristic implemented by the players.

FRASim on the attractiveness function, and overall it allows for a faster convergence on a split of the grid. Finally, we can also see that the interaction between difference in consistency and overlap, as captured by the regression model discussed in Eq 2, behaves as expected. The interaction coefficient, $\beta_4$, for MBIASES is only 0.007, for WSLS rises to 0.016, and goes up to 0.026 for FRA.

**Table 2. Parameters used in simulations shown in Fig 7.**

| Model | Parameters | | | | | | | | | |
|---|---|---|---|---|---|---|---|---|---|---|
| | bias$_{ALL}$ | bias$_{NOTHING}$ | bias$_{LEFT}$ | bias$_{IN}$ | $\alpha$ | $\beta$ | $\gamma$ | $\delta$ | $\epsilon$ | $\zeta$ |
| MBIASES | 0.1 | 0.1 | 0.1 | 0.1 | 0 | 0 | 0 | 0 | 0 | 0 |
| WSLS | 0.1 | 0.1 | 0.1 | 0.1 | 100 | 30 | 31 | 0 | 0 | 0 |
| FRA | 0.1 | 0.1 | 0.1 | 0.1 | 100 | 30 | 31 | 2 | 30 | 0.8 |

## Parameter fit and model recovery

To fit the model parameters to the dataset collected from humans, we constructed a likelihood function on the basis of a multinomial function. The arguments to the latter are the probability vector $(p_{RS}, \ldots, p_{OUT})$ obtained from the model, as well as the vector of observed frequencies $(n_{RS}, \ldots, n_{OUT})$ obtained from the dataset. Both vectors depend on the region a player explored on a round, the score obtained therein, and the area of overlapping tiles with the other player—this is what we called the current situation of the game. The *deviance* of the model with respect to a dataset is obtained by summing the $-2*$log-likelihoods from every situation of the game. We found the optimal parameters that minimize the deviance using the simplex method, implemented in the NMKB function of the DOPTIM package in R.

We performed a model recovery exercise in order to check the accuracy of our parameter estimation method. The results are shown in Fig 8, where we can observe that the method is fairly accurate to fit biases in the case of MBIASES and WSLS (see panels in box (a); first row corresponds to MBIASES; the second to WSLS, and the third to FRA). Moreover, the estimation of $\alpha$ and $\gamma$ (see box (b), top for WSLS and bottom for FRA), is almost completely satisfactory for WSLS. We should observe that parameter fit is sub-optimal in the case of FRA, probably due to the interaction between mechanisms (see bottom rows in boxes (a), (b) and (c)).

## Modeling a shaky hand

It is not uncommon to find players in a fully coordinated dyad falling out of line with respect to their previously selected focal region, and making a small deviation of two or three tiles from this focal region, only to come back to the full region on the next round. The discrepancy between an internal decision and the overt behavior—commonly known as the shaky-hand phenomenon—must be incorporated into our model to allow it to fit the observed behavior from our human subjects. In particular, a shaky hand process allows us to accommodate players' overt tile choices falling outside of a focal region to which they subsequently return.

To get an idea about the extent of this phenomenon, we looked at trials on which DLINDEX$\geq$0.99 and counted the number of misplaced tiles with respect to the closest focal region. We found that the proportion of regions with no misplaced tiles—a proportion we shall refer to as NON-SHAKY—is approximately 0.88.

To account for the shaky-hand phenomenon when fitting our models' parameters to the data, we included an extra layer on top of Eq 3. This additional layer represents the actual behavior, which is theoretically different from the decision made by the player. In this layer, the probability of a focal region is lowered by a factor of NON-SHAKY with respect to the original model according to the following formula:

$$P_{\text{NON-SHAKY}}(k) = \begin{cases} \text{NON} - \text{SHAKY} * P(k), & \text{if } k \text{ is a focal region} \\ 1 - \text{NON} - \text{SHAKY} * (1 - P(k)), & \text{if } k = \text{RS} \end{cases} \tag{6}$$

To derive the formula in the case where $k = $ RS, observe that $\Sigma_{k \in \text{Focals}} P(k) = 1 - P(\text{RS})$. This formula determines the probability vector to be used in the procedure explained above with which we fit the models to the behavioral data.

We also modified simulations to include a shaky hand in the following way. At the end of each round, players choose a region according to Eq 1. However, at the beginning of the round, players choose with probability NON-SHAKY whether a small number of randomly chosen tiles inside and outside the chosen region are flipped (i.e., not visited if in region, or visited if not in region). In this way, the model's choice behavior does not always coincide with the chosen focal region in a proportion approximately equal to $1 - $ NON-SHAKY.

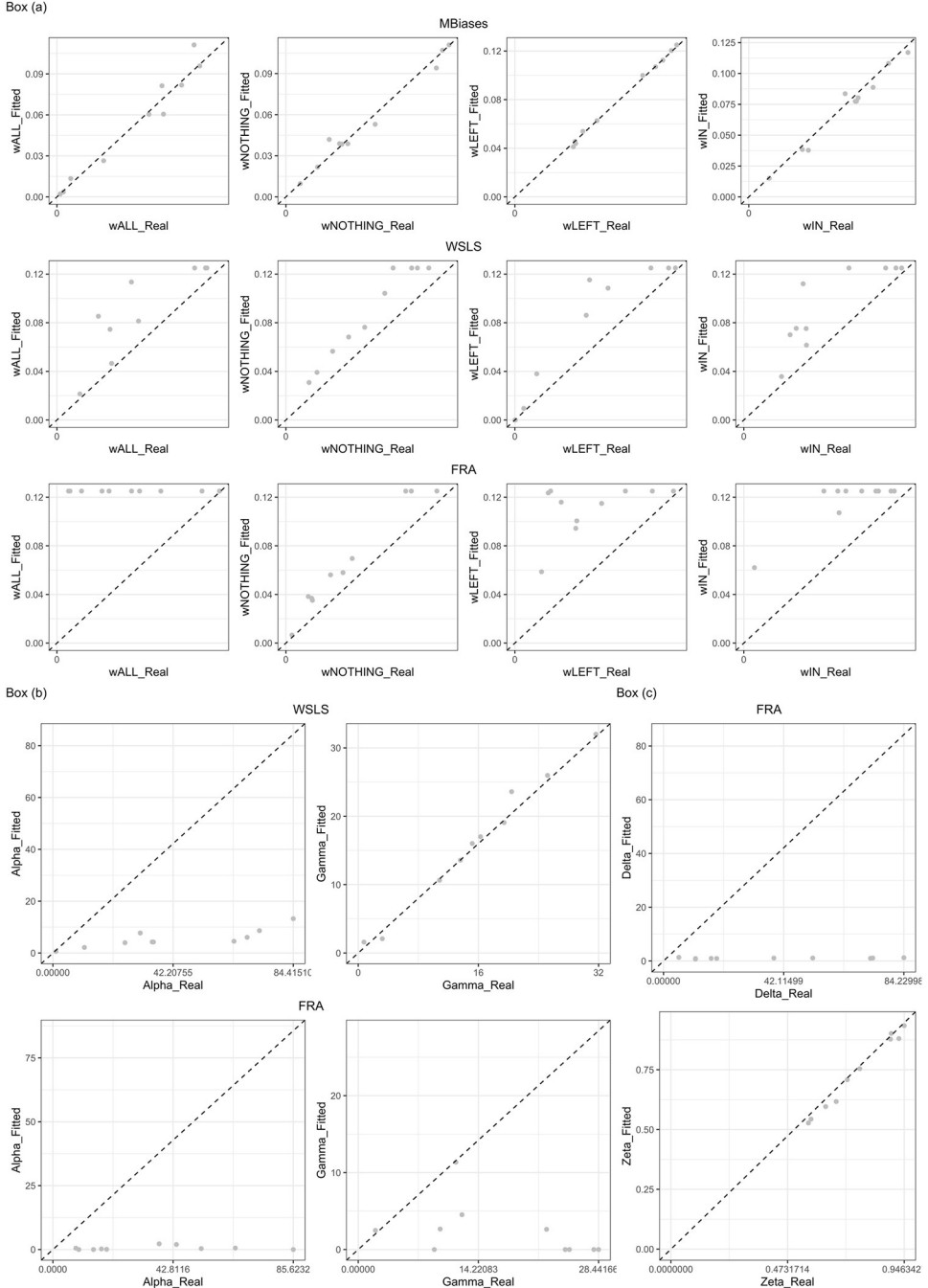

**Fig 8. Model recovery exercise.** All panels plot the value of the parameter that was input to the simulation (*x*-axis) against the parameter recovered by the maximum likelihood method (*y*-axis). Each data point corresponds to one simulation with a given set of randomly chosen parameters. In a perfect world, all points should fall in the *y* = *x* line. Box (a) shows the results for the first four parameters, corresponding to the biases. Box (b) corresponds to parameters controlling "Win Stay, Lose Shift." Box (c) corresponds to parameters controlling FRASim.

## Behavioral data fit

We fit our models to the behavioral data and found their deviance and AIC, summarized in Table 3. Using the Likelihood Ratio Test for nested models, we obtained quantitative evidence

**Table 3. Best parameters, deviance, and AIC for each model.** It also presents the critical value of $\chi^2$ for the Likelihood Ratio Test for nested models.

| Model | Parameters | | | | | | | | | |
|---|---|---|---|---|---|---|---|---|---|---|
| | $\text{bias}_{ALL}$ | $\text{bias}_{NOTHING}$ | $\text{bias}_{LEFT}$ | $\text{bias}_{IN}$ | $\alpha$ | $\beta$ | $\gamma$ | $\delta$ | $\epsilon$ | $\zeta$ |
| MBIASES | 0.13 | 0.076 | 0.058 | 0.005 | 0 | 0 | 0 | 0 | 0 | 0 |
| WSLS | 0.1 | 0.05 | 0.018 | 0.002 | 38 | 30 | 4.6 | 0 | 0 | 0 |
| FRA | 0.06 | 0.05 | 0.003 | 0.000 | 40 | 30 | 15 | 0.5 | 30 | 0.954 |

| Model | Dev. | $\chi^2$ vs. MBIASES | $\chi^2$ vs. WSLS | AIC | ΔAIC vs. MBIASES | ΔAIC vs. WSLS |
|---|---|---|---|---|---|---|
| MBIASES | 1612 | – | – | 1620 | – | – |
| WSLS | 654 | 958* | – | 668 | 952† | – |
| FRA | 557 | 1055** | 97* | 577 | 1043† | 91† |

Conventions: * signifies that $\chi^2$ has 3 d.o.f. and that p < 0.001. Moreover, ** signifies that $\chi^2$ has 6 d.o.f. and that p < 0.001. Finally, † signifies strong support that the winning model is closer to the true generating process over and above the complexity introduced by the extra parameters.

that FRA provides a better account of the underlying choice process and that this model's better fit to the data is not due to over-fitting. Moreover, the AIC shows that there is strong support in favor of FRA with respect to both WSLS and MBIASES, even if we take into account this model's greater complexity, as introduced by the extra parameters.

We simulated 45 dyads playing 60 rounds of the game, for the three models with the fitted parameters from Table 3. In Fig 9 we can see the behavior of these models. In the left panel we can see that FRA and WSLS come quite close to the observed behavior at the dyadic level, and that the former predicts a better coordination than the latter. In the right panel we can see the kernel density estimate of DLINDEX for observed and simulated data. For humans, values of

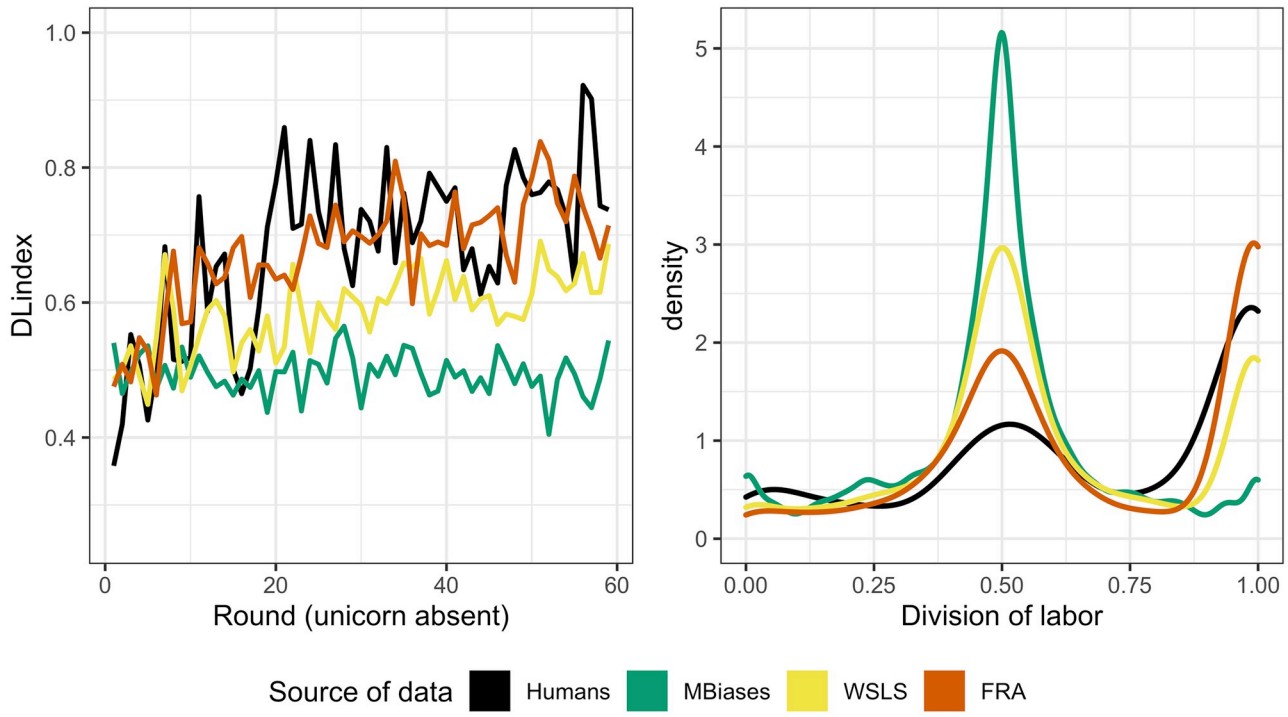

**Fig 9. Behavioral data fit at the dyadic level.** Left: DLINDEX per round, averaged over 45 dyads. Right: Kernel density estimate of DLINDEX.

DLINDEX between 0.5 and 1 are more frequent, representing the fact that humans show more intermediate states of coordination between players as compared to our models' predictions. However, tendencies for both FRA and WSLS are similar to that of humans, with a significantly better fit for the former. To sum up, it seems that WSLS predicts a less efficient division of labor than exhibited by people, whereas FRA and people show a comparable degree of division of labor.

## Discussion

57% of human dyads finished 60 rounds of gameplay with an efficient division of labor. The results from our experiment and our computational models allow us to explain how most dyads managed to split the grid without being able to engage in explicit negotiations. First of all, even though there are $2^{64}$ ways to split the grid, dyads split it in only four different ways. In some sense, these splits are focal points because they have a certain psychological salience [19]. One might have thought that these individual cognitive biases (focal points) would exert an early (in terms of rounds) influence on choices exactly because they are *a priori*, so that agents would have started on Round 1 with strategies of selecting all tiles on the left, top, bottom, or right. If agents understand that these are natural attractors, then through engaging in many levels of iterated thinking based on common knowledge [21], these would be logical starting points. However, players do not generally start with strategies that resemble focal points. Humans are far more idiosyncratic and exploratory in their initial selections of tiles. It is only through repeated interactions that players manifest their *a priori* predispositions/ biases toward certain focal points. In other words, *a priori* biases do not entail that the biases are manifest at the onset of play. It is only through dyadic interaction that these biases are revealed [30].

This is an interesting complement to Schelling's focal points proposal. Schelling's original examples were all one-shot tasks, featuring complex problems (e.g., if you and your wife get lost from one another in the supermarket, in the era prior to cell phones, where will you find each other?). However, Schelling's solution only works if there is just one focal point on which to converge (but can you and your wife think of only one single salient spot in the supermarket?). When there are two or more focal points, further negotiations are required. We propose that, in an iterative task, this implicit negotiation might take place in the form of focal points being revealed after some early attempts, which then attract a player's choices, and the overlap provides hints to the other player to move towards complementary choices.

As for some potential applications of our model outside cognitive science, consider the following. Suppose that humans and bots are cooperating to monitor for infiltrators into a group. They may need to adaptively organize themselves to look on different social media areas for infiltrators. In this case, finding infiltrators is like searching for a unicorn. Taking into account FRA, a bot might want to assess whether a human is close to covering a focal region (e.g., a compact set of forums that cohere based on their topical similarity), and if so move their search to cover complementary regions. Or suppose a situation similar to 'El Farol bar problem' [14], where a group of people is iteratively presented with the choice of taking advantage of a resource, but only a limited amount of people can profit from it per turn. Collaboration may be improved if participants are presented with visual cues about their overlapping turns, which might allow them to move to complementary focal points—i.e., natural and complementary patterns of turns determining when a person is allowed to reach for the resource.

## Conclusion

Interacting individuals, both human and algorithmic, can often arrive at efficient coordinating solutions in a paradigm that incorporates two challenging conditions: Individuals cannot explicitly communicate, and there are multiple coordinating solutions that are initially equally salient. The human and computational results indicate that agents solve this coordination task by beginning with a set of mutually incompatible focal points. Then, via iterated interactions they adjust their behaviors to move toward focal points when they are not at a focal point, stay in a focal point once reached, and shift to a complementary focal point relative to the other player when this other player is close to a focal point. In this way, the coordination that a group forms results from the interplay over time between their *a priori* cognitive biases and the dynamics of their interpersonal interaction [31].

## Data and resources availability

We have made freely available our experimental protocols, datasets, videos, and scripts (in both R and Python) that we used to perform the various statistical tests, plots, model simulation and model checking. We will annotate each of these resources in the following paragraphs.

### Protocols and datasets

**Protocol.** The description of the "Seeking the unicorn" protocol, described in "Materials and methods" section, can be found in the protocols repository https://www.protocols.io/view/seeking-the-unicorn-bts9nnh6. As stated in the main text, we implemented the task using nodeGame, and the code implementing the game is freely available at https://github.com/Slendercoder/Seeking_the_unicorn.

**Datasets.** Using the aforementioned protocol and the nodeGame platform, we obtained data from our human participants. The datasets can be found at the following Open Science Foundation link: https://osf.io/3xcqr/?view_only=162a4ed4834b419ea374c605519c0d1f. They can also be found at the Github repository for the whole project: https://github.com/EAndrade-Lotero/SODCL. A complete explanation of our datasets can be found as a raw text file https://github.com/EAndrade-Lotero/SODCL/blob/master/README.rtf or as a jupyter notebook https://github.com/EAndrade-Lotero/SODCL/blob/master/Dataset_explanation.ipynb. The raw dataset, which only combines and transforms data from multiple json output files from nodeGame can be found in https://github.com/EAndrade-Lotero/SODCL/blob/master/Data/performances.csv. However, the main dataset we used during the analysis, in which we only maintained rows representing rounds with unicorn absent and which includes the relevant measures, can be found in https://github.com/EAndrade-Lotero/SODCL/blob/master/Data/humans_only_absent.csv. The python code used to perform the classification of regions and to obtain the different measures can be accessed here: https://github.com/EAndrade-Lotero/SODCL/blob/master/Python/get_measures.py.

### Videos and graphics of each dyad's behavior

**Videos.** We have reproduced the behavior of each dyad in a video, each frame representing one round of the experiment. In each frame there are two grids, one per player, where black tiles represent tiles covered by only one player and red tiles represent tiles covered by both players. The reproduction of each dyad's gameplay can be accessed at https://github.com/EAndrade-Lotero/SODCL/tree/master/Videos. The reader is suggested to consult videos 140-615, 648-175, and 948-444 for examples of interesting dynamics.

**Graphics with representative measures.** In the following pdf, the reader can consult the representative measures for all dyads, including the score, accumulated score, DL$_{\text{INDEX}}$, consistency and the final magnitude-coded tiles: https://github.com/EAndrade-Lotero/SODCL/blob/master/graphics.pdf. The document was created with the python code in https://github.com/EAndrade-Lotero/SODCL/blob/master/Python/get_graphics.py using the second dataset presented above. This document is an expansion of Fig 2 to all dyads.

## Classification of dyads

The classification of dyads into each of the eight focal regions was done with the help of the R code in https://github.com/EAndrade-Lotero/SODCL/blob/master/R/ClassifyDyads.R. This information was essential for making Fig 3.

## Statistical tests and evidence for qualitative behavior

**Influence of unicorn present.** The R code used to check the influence of the unicorn being present on the player's behavior during the subsequent round with unicorn absent can be found in https://github.com/EAndrade-Lotero/SODCL/blob/master/R/InfluencePresent.R.

**Comparison between one-shot and iterated task.** To compare the behavior observed during the first round to that of all rounds, we used the R code in https://github.com/EAndrade-Lotero/SODCL/blob/master/R/OneShot-vs-Iterated.R, as well as the following python code, which we used to produce the histogram of convergence on a successful division of labor: https://github.com/EAndrade-Lotero/SODCL/blob/master/Python/Histogram.py. This information was essential in making Fig 4.

**Evidence for heuristcs.** The first heuristic we checked against the data was that of "Win Stay, Lose Shift." To analyze the data we used the R code in https://github.com/EAndrade-Lotero/SODCL/blob/master/R/WSLS.R. In the main text we mentioned that this heuristic cannot fully account for an interaction effect between difference in consistency and the amount of overlapping tiles. The R code we used to find this interaction can be found in https://github.com/EAndrade-Lotero/SODCL/blob/master/R/InteractionEffect.R. Another heuristic is that of stubbornness, that is, the tendency of players to reselect tiles when they resemble a focal region. The R code to check this can be found in https://github.com/EAndrade-Lotero/SODCL/blob/master/R/Stubbornness.R. Finally, in the main text we presented an example from an actual gameplay of one of our human dyads to illustrate the FRA heuristic. We have created a video displaying the FRASim measures for some rounds, which can be viewed here: https://github.com/EAndrade-Lotero/SODCL/blob/master/Videos/FRA_in_action.mp4. The gameplay for all rounds can be found in video https://github.com/EAndrade-Lotero/SODCL/blob/master/Videos/435-261.avi. These videos were essential for making Fig 6.

## Simulated data, parameter fit and fitted models

To obtain our simulated data, we implemented the game in a Python code. To run the model, run `>python3 main.py` with the desired parameters. The logic of the game is coded in EmergenceDCL.py, and the functions that properly run the decision process based on the models are in FRA.py. All these files can be freely accessed at https://github.com/EAndrade-Lotero/SODCL/tree/master/Python. In the main text we presented the results of simulated data in Fig 7. The R code used to make this figure can be found in https://github.com/EAndrade-Lotero/SODCL/blob/master/R/Mechanisms.R. The simulated data for the model recovery exercise can be found in https://github.com/EAndrade-Lotero/SODCL/tree/master/Data/Model-Recovery, and the R code to fit the models by means of the maximum likelihood

estimation technique in https://github.com/EAndrade-Lotero/SODCL/blob/master/R/ConfusionMatrix.R. We used this information to create Fig 8 by means of the R code in https://github.com/EAndrade-Lotero/SODCL/blob/master/R/ReadCM.R. To fit the models to human data we used the R code in https://github.com/EAndrade-Lotero/SODCL/blob/master/R/fitModel.R and the information obtained was presented in Table 3. To visualize the models with the parameters fitted to human data, we used the R code in https://github.com/EAndrade-Lotero/SODCL/blob/master/R/plot_fitted_models.R, which produces Fig 9.

## Acknowledgments

We are grateful with the anonymous referees from CogSci2019 and from this journal for valuable comments on earlier versions of this text, and with Evan Nix for running the experimental sessions. EAL is also grateful with Luis Andrade and Carlos Álvarez for useful discussions.

## Author Contributions

**Conceptualization:** Edgar Andrade-Lotero, Robert L. Goldstone.

**Data curation:** Edgar Andrade-Lotero.

**Formal analysis:** Edgar Andrade-Lotero, Robert L. Goldstone.

**Investigation:** Edgar Andrade-Lotero, Robert L. Goldstone.

**Methodology:** Edgar Andrade-Lotero, Robert L. Goldstone.

**Resources:** Robert L. Goldstone.

**Software:** Edgar Andrade-Lotero.

**Visualization:** Edgar Andrade-Lotero.

**Writing – original draft:** Edgar Andrade-Lotero.

**Writing – review & editing:** Robert L. Goldstone.

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
