## [Decision Letter · Decision Letter 0]

3 Mar 2021

PONE-D-21-01872

Self-organized division of cognitive labor

PLOS ONE

Dear Dr. Andrade,

Thank you for submitting your manuscript to PLOS ONE. After careful consideration, we feel that it has merit but does not fully meet PLOS ONE’s publication criteria as it currently stands. Therefore, we invite you to submit a revised version of the manuscript that addresses the points raised during the review process.

The three reviewers have provided constructive and detailed comments. They all agreed that the work has merit. However, there are several major aspects of the paper that need improvements, for which the reviewers have provided constructive suggestions. Please carefully consider them in the revision of your manuscript.

We look forward to receiving your revised manuscript.

Kind regards,

The Anh Han, Ph.D.

Academic Editor

PLOS ONE

Journal Requirements:

Additional Editor Comments:

The three reviewers have provided constructive and detailed comments. They all agreed that the work has merit. However, there are several major aspects of the paper that need improvements, for which the reviewers have provided constructive suggestions. Please carefully consider them in the revision of your manuscript.

Reviewers' comments:

Reviewer's Responses to Questions

**Comments to the Author**

1. Is the manuscript technically sound, and do the data support the conclusions?

Reviewer #1: Yes

Reviewer #2: Yes

2. Has the statistical analysis been performed appropriately and rigorously? 

Reviewer #1: Yes

Reviewer #2: I Don't Know

3. Have the authors made all data underlying the findings in their manuscript fully available?

Reviewer #1: Yes

Reviewer #2: Yes

4. Is the manuscript presented in an intelligible fashion and written in standard English?

Reviewer #1: Yes

Reviewer #2: Yes

5. Review Comments to the Author

Reviewer #1: Overview and general recommendation

This manuscript investigates how humans coordinate their efforts to accomplish a common task by dividing labor. The authors present experimental evidence, through an iterated two-person game, that, although only 57% of the participants managed to divide labor efficiently, successful participants adapt their behavior throughout the different rounds of the experiment and converge to focal points. Finally, the authors propose a series of four models of incremental complexity, and show the “win stay, loose shift”, mirroring of the other player’s actions (i.e., adopting a complementary response) and shaky-hand behaviors are important features of participant’s strategies in the proposed game and may explain how humans coordinate.

I would like to congratulate the authors for a very well written manuscript and very interesting results. The authors have successfully managed to combine empirical and theoretical work to explain a very important question in human behavior, i.e., how humans coordinate in the absence of (explicit) communication. For this reason, I believe this paper should be accepted. Nevertheless, I do have a number of minor comments that I would like to have addressed, before the paper is published.

Minor comments

Abstract:

The abstract should contain an indication of the results that are being presented in the paper. Additionally, the authors mention that this work has several potential applications. Yet, this is never discussed in the main article, so the reader is left with the question of how are these findings really applicable to multi-agent systems or human-robot collaboration? I believe the authors should add this to the discussion.

Introduction:

The authors indicate that “maximization of expected utility often is not sufficient to explain why individuals act in accord with one particular Nash Equilibrium instead of another”. Although I do agree that ‘bounded rationality’ often accounts for human behavior better than a fully rational model, this does not mean that humans are not optimizing. Rather, they are optimizing with constrained resources, and indeed may adopt heuristics. Moreover, one has to be careful, since game equilibrium may be altered both by player’s beliefs (in case that rationality is not assumed) and preferences, which affect utility.

In this line, the authors make a strong claim by saying that “game-theoretic explanations might only provide distal causes of behavior”. Game theory is a large field, and not all of it is about studying Nash equilibria. In fact in Evolutionary Game Theory, no assumptions are made about rationality, nor ‘bounded rationality’ (see ref. [1-4] for evolutionary dynamics on coordination games). I thus believe, the authors should relax their claim.

Experimental design:

From the experimental description, it is not clear whether participants receive an incentive (in this case the course credits) in function of their performance during the game. If this is not the case, I do not understand how one can make the argument that maximization of utility is not relevant, since this is simply not being tested in the experiment.

Moreover, I would like the authors to clarify whether the students that participated in the experiment belong to some course the authors (or their colleagues) teach. In which case, I would argue that there may be some confounding effects due to players knowing each other.

Finally, this effect might be intensified since some of the sessions had a very low number of dyads (2 or 3) which makes it easier for participants to identify with whom they are playing, and indirect reciprocity might play a role in the results.

I do not have an issue with the results of the experiment being published, and fully understand that not all behavioral experiments follow the precepts of behavioral economics. Yet, the authors should be more critical in the discussion and comment on these possible limitations of the study, as well as give an overview of the general applicability of their conclusions.

Regression on Figure 5:

The regression shown on Figure 5 does not seem to fit well the data (there are many points far away from the mean). This may indicate that a linear regression is not appropriate in this case. Moreover, although the authors indicate that their data shows that the interaction between consistency and overlapping tiles is positive, this is only true assuming the model they are regressing. If they regressed only over the variables diffConsist(n) * Overlap(n-1) the result might be different. Thus, this should be indicated in the manuscript.

Figure 7:

A similar problem occurs in Figure 7 top-right panel. It is difficult to extract any meaning from that figure, but it appears that a logistic model would be a better fit to that type of data. With so much variance in the points, the linear fit may not be giving you proper information.

it would also be interesting to find out how players react to the unicorn not existing. This could be done by cheeking the effect on the player's behavior in subsequent rounds when the unicorn isn't found and when it is found.

Finally, I only found one typo, which really shows the effort the authors have put in writing the manuscript:

Page 10-line 1: “mayority” should be “majority”

There is also a formatting error in Ref. [15].

References

[1] Zhang, Chunyan, et al. "Evolutionary dynamics in division of labor games on cycle networks." European Journal of Control 53 (2020): 1-9.

[2] Stephenson, Daniel. "Coordination and evolutionary dynamics: When are evolutionary models reliable?." Games and Economic Behavior 113 (2019): 381-395.

[3] Pacheco, Jorge M., et al. "Evolutionary dynamics of collective action in N-person stag hunt dilemmas." Proceedings of the Royal Society B: Biological Sciences 276.1655 (2009): 315-321.

[4] Ohtsuki, Hisashi. "Evolutionary dynamics of coordinated cooperation." Frontiers in Ecology and Evolution 6 (2018): 62.

Reviewer #2: This manuscript investigates the cognitive processes behind the emergence of division of labour in human groups, without the use of explicit communication between individuals. It addresses this using a novel task involving uncovering squares on a 8x8 grid, where dyads must try to minimise overlap while maximising coverage. Experiments are performed with human subjects, and a computational model is developed and fitted to the data.

Overall the results look sound to me. I do think that the task is quite complicated. I wonder whether the same results would be obtained in a simpler game with only two focus points? Could the finding that players do not immediately use their heuristic biases, but take a few rounds to start using these, be simply an artefact of the complexity of the task and the number of different focal points available? I would like to see this discussed, as well as some discussion to justify why a simpler task isn't used. For example, you could even just use a smaller grid? What made you choose an 8x8 grid?

The abstract mentions potential applications outside of cognitive science, but these are never discussed in the manuscript. The discussion section should be expanded to discuss each of the suggested applications in turn, or alternatively, the mention of these applications should be removed.

The first two sentences of the introduction are saying the same thing -- that an individual can benefit from DoL. I think you mean that there are both group and individual benefits, i.e. it's not altruistic.

The third paragraph on game theory is quite difficult to follow. It would be good to give an example of what a payoff matrix might look like, or at least a concrete example of how the payoffs depend on the choice of actions of both the focal player and others.

Could you give some explicit examples of real world examples that are modelled by the unicorn game? How common is division of labour in humans with no explicit communication? You should give more empirical examples to better motivate the work. Do you expect your results to generalise to n-person interactions (which are surely more common in human groups)?

6. PLOS authors have the option to publish the peer review history of their article (what does this mean?). If published, this will include your full peer review and any attached files.

Reviewer #1: No

Reviewer #2: No

---

## [Author Response · Author response to Decision Letter 0]

9 Apr 2021

Response to academic editor’s points:

- Laboratory protocols were suggested to be deposited in protocols.io. Our protocols were uploaded to said repository and the link was added to the ‘Materials and methods’ section in the paper.

- Additional details regarding participant consent were requested. The ‘Participants and procedure’ section in the manuscript was augmented to make explicit that informed consent was requested from participants before they could proceed to the test and that no minors were involved in it.

- The grant information was fixed to match ‘Funding Information’ and the financial disclosure.

Response to reviewer 1’s comments:

- The abstract now contains an indication of the main result.

- We added to the ‘Discussion’ section some suggestions on how our findings are applicable to human-robot collaboration and multi-agent systems, and a paragraph in the ‘The task’ section to show how we can model some real-world situations.

- The reviewer rightly pointed out that our claim that game-theoretic explanations provide only distal causes of behavior is too strong. We have relaxed our statement by appealing to Lieder’s and Griffiths’ “Resource-rational analysis” paradigm. See the ‘Introduction’ section for more details.

- Relaxing the above mentioned claim also addresses the reviewer’s point that our experiment cannot say anything about maximization of utility being relevant or not, since our subjects did not receive incentives for their performance.

- The Participants and procedure section has been updated with a more complete description of the students that participated in the experiment, who were NOT from either of our classes, but from the standing human subject pool at Indiana University. 

- The reviewer raised an issue with the regressions in Figure 5 top left and Figure 7 top right. Both of these regressions feature scatter plots of consistency on round n vs. score on round n-1 (the former for behavioral data and the latter for simulated data). It is correct to claim that a linear model is not the right model to fit in this case. For one thing, the points are too far away from the means, there is heteroscedasticity and too much variance. However, we are not fitting a model for predictive purposes. The figures are just intended to give a visual way to present a tendency in the qualitative behavior. Tendencies – viz Win Stay, Lose Shift – can be appreciated from the respective Pearson correlations, which correspond to the slants of the linear regressions. We included “hedging” phrases in the manuscript to warn the reader about this particular use of linear regressions and included the Pearson correlations. We would also like to point out that a logistic regression is no better an option than linear regression here because the outcome variable (consistency on round n) takes ordered values.

- As for the interaction coefficient in the regression DLindex ~ Consistency + DifConsistency*Overlap, the reviewer pointed out that the result might be different if Consistency was not included. However, the interaction coefficient is even higher and significant when this contribution is excluded (β_3≈0.021;p-value<0.001; observe that the interaction coefficient is now β_3, not β_4). But we think that the effect of consistency should be taken into consideration to isolate the interaction effect, since consistency already has quite an important contribution to DLindex. We added a phrase in the manuscript to point out that the sign of the contribution doesn’t change if we leave out the contribution of consistency. 

- Concerning the question of how players react to the unicorn being present or not, we have modified the second paragraph in the ‘Results’ section to address this question. We show that rounds with unicorn present made players more active, in the sense that they explored a larger portion of the grid on the next round.

- The typo on page 10 and the formatting error in the references were corrected.

Response to reviewer 2’s comments:

- The reviewer rightly points out that the task is quite complicated. This is a point to be taken into account for future experiments. However, we justify our task on the basis of our objective to determine how one player adapts to the other. To this end, we resorted to an incremental task of uncovering tiles one by one. Moreover, earlier pilot runs showed that a smaller grid didn’t invite players to divide it between them, and that a more demanding task was more likely to be divided. We adapted the second to last paragraph of ‘The task’ section to address this issue, and the reviewer’s query as to why we chose the grid size that we did.

- The reviewer is concerned that, in our task, the need for iterations to achieve coordination may be “an artefact of the complexity of the task and the number of different focal points available”. After careful consideration we think that this is not a problem, but an important claim we want to make. Schelling’s original examples were all one-shot tasks, featuring complex problems (e.g., if you and your wife get lost from one another in the supermarket, in the era prior to cell phones, where will you find each other?) but the focal point solution only works if there is just one focal point (can you and your wife think of only one single salient spot in the supermarket?). The fact is that two or more focal points call for further negotiations. It is no surprise that solving a task with two or more focal points, just as our task, requires iterations to achieve coordination. We don’t see this as a shortcoming, but rather addressing the issue at hand, namely, how is this implicit negotiation achieved? We modified the fifth paragraph in the ‘Results’ section where this issue is brought up and included a paragraph in the ‘Discussion’ section to make explicit that multiple focal points require further negotiations and how our proposal addresses this issue.

- We added to the ‘Discussion’ section some suggestions about how our findings are applicable to multi-agent systems and human-robot collaboration.

- We fixed the second sentence of the introduction to avoid repeating ourselves.

- In the third paragraph of the ‘Introduction’ section, we have included an example of a (partial) payoff matrix to make the presentation easier to follow.

- We have provided in the ‘The task’ section a number of real-world situations that can be modeled by our task to better motivate it. The generalization to an n-person interaction task known as the `El Farol bar problem’ is mentioned in the ‘Discussion’ section as a possible application of our FRA model.

---

## [Decision Letter · Decision Letter 1]

26 May 2021

PONE-D-21-01872R1

Self-organized division of cognitive labor

PLOS ONE

Dear Dr. Andrade,

Thank you for submitting your manuscript to PLOS ONE. After careful consideration, we feel that it has merit but does not fully meet PLOS ONE’s publication criteria as it currently stands. Therefore, we invite you to submit a revised version of the manuscript that addresses the points raised during the review process.

ACADEMIC EDITOR: The two reviewers have provided constructive and detailed comments. They both agreed that the work is interesting, relevant and would provide a good contribution. However, there are some aspects of the paper that need improvements, for which the reviewers have provided constructive suggestions. Please carefully consider them in the revision of your manuscript.

We look forward to receiving your revised manuscript.

Kind regards,

The Anh Han, Ph.D.

Academic Editor

PLOS ONE

Journal Requirements:

Additional Editor Comments (if provided):

The two reviewers have provided constructive and detailed comments. They both agreed that the work is interesting, relevant and would provide a good contribution. However, there are some aspects of the paper that need improvements, for which the reviewers have provided constructive suggestions. Please carefully consider them in the revision of your manuscript.

Reviewers' comments:

Reviewer's Responses to Questions

**Comments to the Author**

1. If the authors have adequately addressed your comments raised in a previous round of review and you feel that this manuscript is now acceptable for publication, you may indicate that here to bypass the “Comments to the Author” section, enter your conflict of interest statement in the “Confidential to Editor” section, and submit your "Accept" recommendation.

Reviewer #1: All comments have been addressed

Reviewer #2: All comments have been addressed

2. Is the manuscript technically sound, and do the data support the conclusions?

Reviewer #1: Yes

Reviewer #2: Yes

3. Has the statistical analysis been performed appropriately and rigorously? 

Reviewer #1: Yes

Reviewer #2: I Don't Know

4. Have the authors made all data underlying the findings in their manuscript fully available?

Reviewer #1: Yes

Reviewer #2: Yes

5. Is the manuscript presented in an intelligible fashion and written in standard English?

Reviewer #1: Yes

Reviewer #2: Yes

6. Review Comments to the Author

Reviewer #1: I would like to congratulate the authors for a great work in adapting the manuscript, and I recommend their work for acceptance. Yet, I do still have some minor comments:

There are still some typos in the text (e.g., Line 259: Fi 5 instead of Fig. 5), so I recommend that the authors review it before publication.

The authors indicate that a logistic regression would not work in their setting since “the outcome variable (consistency on round n) takes ordered values”. However, for these cases you may use an Ordinal Logistic Regression (see https://stats.idre.ucla.edu/r/dae/ordinal-logistic-regression/ ).

Finally, the availability of resources including Code and Data is spread over the document, which makes it sometimes difficult (or cumbersome) to find/understand this information. Therefore, I recommend that the authors add an extra session at the end of the manuscript titled, e.g., “Data and resources availability”, and group all the links to code and data there with some explanation.

Additionally, I might have missed it, but I could not find any direct link to the data of the experiment. What I did find was this CSV file in the authors Github: https://github.com/EAndrade-Lotero/SODCL/blob/master/Data/humans_full.csv . Is this the data of the experiment? If this is the case, I recommend that the authors add a README.md file to this folder, explaining the format of the datasets contained in it, and the meaning of each column.

Optionally, I do think it is best to deposit the data in a dataserver, e.g., https://datadryad.org/stash, where information about the data is correctly stored.

Reviewer #2: (No Response)

7. PLOS authors have the option to publish the peer review history of their article (what does this mean?). If published, this will include your full peer review and any attached files.

Reviewer #1: No

Reviewer #2: No

---

## [Author Response · Author response to Decision Letter 1]

9 Jun 2021

Response to Reviewers

Response to reviewer 1’s comments:

• The reviewer was concerned about the typos in the text. We made a thorough review to fix as many of them as possible.

• In the previous versions of the manuscript, we used a linear regression to explore the relationship between score on round n-1 and consistency on round n. The reviewer was concerned that this was not an adequate model of the data and suggested we should use an ordinal logistic regression instead. To implement this analysis, we partitioned both variables into three intervals, so that the analysis could be carried out on two ordinal variables. We explored the relationship between the levels of these variables by means of an ordinal logistic regression. We showed that the results favor a positive relationship from low scores to high scores suggesting that trials where players received a high score elicited a more consistent behavior on the next round as compared to trials where players received low or medium scores. We visualized this relationship by means of a comparison of the boxplots of consistency for the three levels of score. This plot now replaces the original linear regression in Figure 5. A similar analysis was conducted with the simulated data and the regression in Figure 7.

• We added an extra section at the end of the manuscript in which we groupe the explanation of all our data and code. We provided a brief explanation of each resource and provided links to their respective repositories.

• Our datasets are improved with a detailed explanation in a README file (as well as a jupyter notebook) and are located not only in the main Github repository, but also in the Open Science Foundation website (https://osf.io). Links to the datasets and explanations are added to the “Data and resources availability” section. We decided against the Datadryad website to store our datasets because at this moment we cannot afford their fee.

---

## [Decision Letter · Decision Letter 2]

29 Jun 2021

Self-organized division of cognitive labor

PONE-D-21-01872R2

Dear Dr. Andrade,

We’re pleased to inform you that your manuscript has been judged scientifically suitable for publication and will be formally accepted for publication once it meets all outstanding technical requirements.

Kind regards,

The Anh Han, Ph.D.

Academic Editor

PLOS ONE

Additional Editor Comments (optional):

Reviewers' comments:

Reviewer's Responses to Questions

**Comments to the Author**

1. If the authors have adequately addressed your comments raised in a previous round of review and you feel that this manuscript is now acceptable for publication, you may indicate that here to bypass the “Comments to the Author” section, enter your conflict of interest statement in the “Confidential to Editor” section, and submit your "Accept" recommendation.

Reviewer #1: All comments have been addressed

2. Is the manuscript technically sound, and do the data support the conclusions?

Reviewer #1: Yes

3. Has the statistical analysis been performed appropriately and rigorously? 

Reviewer #1: Yes

4. Have the authors made all data underlying the findings in their manuscript fully available?

Reviewer #1: Yes

5. Is the manuscript presented in an intelligible fashion and written in standard English?

Reviewer #1: Yes

6. Review Comments to the Author

Reviewer #1: (No Response)

7. PLOS authors have the option to publish the peer review history of their article (what does this mean?). If published, this will include your full peer review and any attached files.

Reviewer #1: No

---

## [Editor Report · Acceptance letter]

5 Jul 2021

PONE-D-21-01872R2 

Self-organized division of cognitive labor  

Dear Dr. Andrade-Lotero:

I'm pleased to inform you that your manuscript has been deemed suitable for publication in PLOS ONE. Congratulations! Your manuscript is now with our production department. 

Kind regards, 

on behalf of

Dr. The Anh Han 

Academic Editor

PLOS ONE